# Communities' perceptions towards cervical cancer and its screening in Wolaita zone, southern Ethiopia: A qualitative study

Birhanu Wondimeneh Demissie[1]☯, Gedion Asnake Azeze[1]☯*, Netsanet Abera Asseffa[2], Eyasu Alem Lake[1], Befekadu Bekele Besha[1], Kelemu Abebe Gelaw[1], Taklu Marama Mokonnon[1], Natnael Atnafu Gebeyehu[1], Mohammed Suleiman Obsa[1]

1 College of Health Sciences and Medicine, Wolaita Sodo University, Wolaita Sodo, Ethiopia, 2 School of Public Health, College of Medicine and Health Sciences, Hawassa University, Hawassa, Ethiopia

☯ These authors contributed equally to this work.
* dearged2011@gmail.com

## Abstract

### Background

Cervical cancer is a malignant neoplasm from cells originating in the cervix uteri. Any woman who is sexually active is at risk of getting HPV. Women in sub-Saharan Africa region have higher chance of developing the disease. There are nearly 26 million Ethiopian women who are over the age of 15 and believed to be at risk of getting HPV. Regrettably, Ethiopian women typically present for cervical cancer care at a late stage in the disease, where treatment is most ineffective.

### Objectives

To explore communities' perceptions of cervical cancer and screening among women in Wolaita zone, southern Ethiopia.

### Methods

A qualitative research using focused group discussions and in-depth interviews was used to explore communities' perceptions of cervical cancer and screening among women in Wolaita zone, southern Ethiopia from March 2018-November 2019. The study participants were men, women and communities who were residents of the study settings and were not health professionals. All focused group discussions (FGDs) and key informant interviews were transcribed and entered into Microsoft Word and thematic content analysis was done.

### Results

A total of fifty-nine participants participated in both FGD (three with men and six with women) and in-depth interviews (IDIs). Most participants have not heard about cervical cancer but know cancer in general. Participants mentioned that the disease usually relates to many births and unprotected sexual intercourse but none mentioned HPV infection. Most of the participants perceive that cervical cancer is incurable and assume that it could be prevented but they think they are not vulnerable to the disease and screening is not necessary.

**Data Availability Statement:** All relevant data are within the paper and its Supporting Information files

**Funding:** The author(s) received no specific funding for this work.

**Competing interests:** The authors have declared that no competing interests exist.

## Conclusion

This study indicates that rural communities in the zone had limited knowledge about cervical cancer and even less about risk factors, screening, treatment and prevention. There is a great need for cancer education and prevention in Ethiopia.

## Background

Cancers that originate in the female reproductive system are referred to as women's reproductive cancers [1]. Cervical cancer is a malignant neoplasm from cells originating in the cervix uteri. It is one of the greatest threats to women's lives [2]. A sexually transmitted virus called Human Papilloma Virus (HPV) is responsible for more than 99% of cervical cancer cases and its precursors [3]. Any sexually active woman, including those with only one partner, is at risk of getting HPV at some point in their lives. People infected with HPV could transmit the virus even when they are asymptomatic [4].

Globally, an estimated 311,000 women worldwide died from cervical cancer in 2018; indeed, 85% of the global burden occurs in low- and middle-income countries that lack organized screening and HPV vaccination programs. Moreover, it is the third most common cancer worldwide and the leading cause of cancer deaths in many countries in Africa [5–8]. Besides, it has been projected the incidence of cervical cancer could increase by 60% in the next 20 years [9]; in a sense that cervical cancer could kill more women than maternal mortality in the next few decades. Cervical cancer rates vary worldwide, with the lowest incidence in developed countries of the European region such as Finland, Switzerland and Greece [10]. On the other hand, it is more prevalent in African countries, including Swaziland, Malawi, Zambia, and Zimbabwe [11, 12]. The effectiveness of population-based cervical cancer screening has been reflected by sharp declines in cervical cancer incidence in high-income countries [13].

Cervical cancer progresses slowly from precancer stage to invasive cancer. It is entirely curable if detected early with effective screening; however, service accessibility and community awareness play a significant role in screening more women [14]. Unfortunately, studies have shown that women aged 25 to 64 years in low- and middle-income countries have the least screening rate, only 3.5% in a given three-year period [7]. For instance, a recent study conducted in Eastern Uganda showed only 4.8% of females aged between 25 and 49 years had ever been screened for cervical cancer. The study further reported barriers to accessing screening services, including lack of awareness, negative individual perceptions, and health facility related challenges [6]. Another study from Ethiopia has identified various inter-related barriers and challenges for early health-seeking behaviour for cervical cancer, such as the insidious nature of the disease, individual-level factors, community-level factors, and institutional-level factors [15].

In 2018, cervical cancer was the second leading cause of cancer deaths in Ethiopia for women aged 15 to 44 years [16]. Evidence also shows that nearly 27.19 million Ethiopian women over the age of 15 are believed to be at risk of developing cervical cancer [17]. However, this figure could be low given the nation has no cancer register center. Besides, cervical cancer screening and treatment service is either unavailable or has a considerably diminished chance of success [1, 9]. Regrettably, Ethiopian women typically present for cervical cancer care at a late stage in the disease, where treatment is most ineffective [14]. The reasons for this are not yet established and need to be studied. In addition, to the best of our understanding,

no study has yet explored communities' perceptions of cervical cancer and screening in Southern Ethiopia, specifically the Wolaita zone. Therefore, findings from this study are expected to contribute to policy and program strategies to better address the cervical cancer needs of women in the community. Ultimately, the study will significantly contribute to implementations that reduce cervical cancer-related deaths in the southern part of Ethiopia and the country as a whole.

## Methods and materials

### Study setting and period

This study was conducted in Wolaita Zone, Southern Ethiopia, located 328kms south of the capital city, Addis Ababa, from March 2018 to November 2019. The zone has an estimated 2.7million population, among the densely populated zones in Ethiopia. For administrative purposes, the Wolaita zone is divided into 15 districts. This study was conducted in two districts of Wolaita Zone, namely Boloso Sore and Sodo Zuriya districts in two urban and two rural *Kebeles* (the lowest administrative unit in Ethiopia). Boloso Sore district had 29 *Kebeles*, of which 25 are rural and the rest four are urban *kebeles*, whereas Sodo Zuriya had 36 *Kebeles* with five urban and 31 rural *Kebeles*.

### Study design

A qualitative descriptive study design using focused group discussions and in-depth interviews was used to explore communities' perceptions of cervical cancer and screening among women in Wolaita zone, southern Ethiopia.

### Participants

The study participants were men and women who were residents of the study settings and were not health professionals. They were recruited through purposive sampling, and recruitment was done through the health extension workers. Inclusion criteria consisted of: (1) 18 years of age and older; (2) educated and uneducated, traditional birth attendants, religious leaders, and women development armies. Exclusion criteria consisted of: (1) prior history of cervical cancer; (2) those who had any training on cervical cancer and (3) currently on the treatment of cervical cancer. A proportional number of FGDs and IDIs were held in the two selected districts.

### Sampling size and sampling technique

This study was conducted in the two randomly selected districts of Wolaita Zone, namely Boloso Sore and Sodo Zuriya districts; it was conducted in two urban and two rural *Kebeles*. Two (one urban and one rural) *kebeles* were randomly selected from each woreda (district). We used purposive sampling from each of the selected kebeles to get the study participants who would be willing to share their experience with cervical cancer. Nine FGDs (a total of 51 participants) and eight individual IDIs were held in each of the four selected *kebeles*. Thus, each FGD had 5–7 participants. The saturation of information determined the number of FGDs and IDIs.

### Data collection tools

Interview guides were used to explore communities' perceptions of cervical cancer and screening (S1 File). The tools were unstructured except for the background information.

## Data collection procedure

Before the Focus Group Discussions and in-depth interviews, research assistants (four female and two male nurses) were trained on how to conduct the interview. The research assistants had a Bachelor of Science (BSc.) degree in Clinical Nursing. They were recruited from Wolaita Sodo University Teaching and Referral Hospital based on their experience on prior qualitative research data collection. Research assistants were fluent in the required local languages and they used the general discussion guide to prompt discussion and elicit further details through probes. Health extension workers of the respective *kebeles* identified the participants. The FGDs and IDIs were conducted in a place convenient for the participants that are quiet and close to their home, except for the Sodo Zuria participants of the women's development army, who preferred health centres. The Boloso Sore participants were interviewed in Wolaitigna, and the Sodo participants were interviewed in Amharic. The participants were divided into four different groups based on their age, because in the local communities of Wolaita, young people may shy away from speaking if included in the group of older adults and vice versa.

During FGDs, there were two research assistants, a facilitator and a note taker, but one interviewer did the IDIs. The moderator gave codes to each participants before the start of the discussion. The codes were used as a name during the conversation, transcription, and translation as the two members of the team supervised information gathering.

**Data quality assurance.** The unstructured questionnaires were prepared in English, translated into Amharic and Wolaitigna, and then returned to English to check for consistency. Before actual data collection, two days of training was given to research assistants on tools and procedures of conducting qualitative research and interviewing sensitive topics. The IDIs and FGD guide was revised prior to administration during pre-testing on six individuals and a group of two, respectively with Wolaitegna and Amharic speakers who shared similar demographic characteristics to the study participants but did not participate in the actual data collection. The IDIs took place at the respondents' houses in private and confidential settings.

**Data management.** The investigators chose and followed clear file naming, developed a data tracking system, established and document data transcription/translation procedures and proved quality control procedures.

**Data analysis.** All FGDs and key informant interviews were transcribed and translated verbatim from the local language to English by individuals fluent in both languages. The transcripts were entered into Microsoft Word and thematic content analysis was done. Specifically, the coding process involved identifying major themes in each of the transcripts. Identified themes were compared across the transcripts to determine differences and similarities in the perspectives of the study. Three themes and six subthemes were identified: awareness about cervical cancer, risk factors for cervical cancer, awareness about symptoms of cervical cancer, perceptions about cervical cancer screening, and prevention and treatment of cervical cancer. Audio files, transcripts, and informed consent forms were stored in password-protected files.

**Ethical clearance.** Participants were briefed on the purpose and objective of the research. We also assured them of the confidentiality of the information. Permission was also requested and any further move was made on their approval. Wolaita Sodo University gave letter indicating the area of research. Ethical approval was obtained from the Ethical Review Board of College of Health Science and Medicine, Wolaita Sodo University. Letters of support were obtained from the relevant authorities of all quarters from where the data were collected. The individual in this manuscript has given written informed consent (as outlined in the PLOS consent form) to publish these case details.

## Results

### Characteristics of participants

A total of fifty-nine participants participated in both FGD (three with men and six with women) and IDIs. The average age (mean ± standard deviation) was 38 ± 5 years. The study participants were primarily married (79.7%), with 59.3% reporting having less than a secondary level education (Table 1).

### Awareness of cervical cancer

Regarding cervical cancer awareness, most participants have not heard about cervical cancer but know cancer in general mainly heard from health extension workers. Those who knew cervical cancer mentioned it as "*Barka* [women's disease name in Wolaita] *that it is related to fistula*" and *"Yemahitsen ber nekersa"* [in Amharic] (34 years old female, Sodo Zuria).

Among the FGD participants, there was a misconception about what cervical cancer is. Seven of the FGD participant groups associated cervical cancer with death. One participant mentioned, *"Anything that has the name cancer in it is deadly regardless of the type and no cure to it. It is a disease that rots, spoils, makes you thin and kills"* (31 years old male, Sodo Zuria).

On six IDIs, mothers mentioned that cervical cancer is a disease that misplaces blood veins, stating, *"cervical cancer misplaces your blood veins particularly veins of your womb. . .{mahitsani barqatis-barqa} [in Wolaita language]. Traditional healer needs to tie womb {yelokota} and will instruct foods to eat."* (37 years old female, Boloso Sore).

### Risk factors for cervical cancer

Except for one FGD and three IDI participants, others mentioned that the disease usually relates to many births and unprotected sexual intercourse, but none mentioned HPV infection. A woman who had similar symptoms to a cervical cancer stated that, *"giving many births causes such symptoms, not the one you call [cervical cancer]. I also heard that cervical cancer could be caused by unrestrained sexual intercourse"* (40 years old female, Sodo Zuria). One participant mentioned that poor personal hygiene and how we touch our cervix determine

**Table 1. Socio-demographic characteristics of participants, Wolaita Zone, Southern Ethiopia, 2019.**

| Characteristics of participants | Values |
|---|---|
| **Sex** | |
| Male | 17 |
| Female | 42 |
| **Age** | |
| <30 | 8 |
| 30–40 | 23 |
| 41–50 | 16 |
| >50 | 12 |
| **Marital Status** | |
| Married | 47 |
| Divorced | 7 |
| Widowed | 5 |
| **Educational Background** | |
| No formal education | 24 |
| Primary | 11 |
| Secondary | 16 |
| Above secondary | 8 |

whether we get cervical cancer. She stated, "*cervical cancer is caused by rarely washing our private parts and touching cervix by napkins; we should be washing only by using water*" (29 years old female, Boloso Sore).

The other cause mentioned by two of the women FGDs and two IDIs was expulsion womb. One participant said "*During childbirth, most women don't go to a health facility, and that leads to prolonged labour which eventually exposes them to the expulsion of the womb which leads to cervical cancer*" (36 years old female, Boloso Sore). The other participant supported the idea and described, "*we the rural women have many things to do, such as bearing and rearing children, looking after domestic animals, sometimes farming, cooking foods, etc. Excessive pressure from such routine tasks may lead to womb expulsion which causes cervical cancer*" (31 years old female, Sodo Zuria).

Among all the men FGD participants, they stated that some harmful traditional practices are stated as the risk factor for cervical cancer. One FGD participant stated, "*when a young girl who is under 18 years gets married to an older man that might expose her to cervical cancer*" (36 years old male, Sodo Zuria).

## Awareness of the symptoms of cervical cancer

In IDIs, only four IDI participants were aware of the symptoms of cervical cancer, and the rest were not even familiar with the term 'cervical cancer'. "*I just heard the disease you called [cervical cancer] here, let alone knowing the symptoms*" (41 years old female, Sodo Zuria).

When we informed them of the symptoms of cervical cancer, the participants reported regarding their friends who had died from cervical cancer years ago.

> "*Now I knew that our friends who had foul-smelling, frequent bleeding from her private parts and pelvic pain were suffering from this disease-poor they. None of us knew that it was cervical cancer; they spent months alone. For years, many of our families and neighbours have died from this disease, and we are just beginning to learn about it now. One of my old aunties who suffered from this disease had an odour that comes from her womb would not let you sit around her, so she was stigmatized, assuming that it was HIV. We were not able to be around her, let alone others. She used to say I should have died during my childhood*" (33 years old female, Boloso Sore).

In FGDs, male participants associated the symptoms with a curse and other diseases, such as Acquired Immune Deficiency Syndrome (AIDS), and some argued that the disease with such symptoms itself is AIDS.

> "*The symptoms you just told us are also symptoms we see in AIDS patients, perhaps it is AIDS-related one*" (40 years old male, Boloso Sore).

## Perceptions regarding cervical cancer screening

Regarding cervical cancer screening, most of the participants were not aware of the availability of such services, and no participant knew that screening is performed for the sake of prevention. In FGDs, most participants explained that one needs to visit the health facility when they experience some discomforts or present with obvious illnesses, such as headache and abdominal pain, amongst others.

> "*How do we go to the health facility when the disease does not show any symptoms; we have to be sick to be seen by a health professional*" (35 years old female, Boloso Sore). One FGD

participant mentioned that *"what is the point of screening? After all, cancer is a killer; better off not knowing cancer will kill you"* (34 years old male, Sodo Zuria).

The need for screening was explained to participants. Although most believed that going for the screening was a good idea, some participants expressed concern about the procedure, having to be done vaginal exam using a speculum. One FGD participant who was 38 years old expressed her fear at which many others nodded their head approving her statement.

*"[tenayi edime gitta. . ..wolayi bolcho.] I am an older woman who is no longer giving birth. How do I lie in bed on my back like that and show my private part to a young girl or male at healthcare facility? Hereafter, my body is an honor for me that I should not reveal to anyone. It is an honor!"* (38 years old female, Sodo Zuria).

In another IDI, a woman expressed her concern that inserting metal in a body is unsafe, perhaps worse. *"What do inserting metals do in a woman who did not show any symptoms of diseases; perhaps the metal will cause another disease condition that was not there in the first place"* (29 years old female, Sodo Zuria).

On the other hand, a woman who had the chance to be screened at the health center told her experience by saying, *"Once I had pain all over my body and went to a health center. The person* [health professional] *respectfully treated and prescribed the drug. Still, in the end, he requested me to go to number 13 [*cervical cancer screening room*], where a woman counselled me for screening, but the moment she instructed me to put off my clothes and lie on a high bed. I was scared! I told her that I felt better and got drugs for what I came for, and then I left the room. When I told my friends how they do the screening, they stated they would never get that kind of screening. The service should not be that way; we should be served for what we go for"* (40 years old female, Boloso Sore). Participants believed that the decision to be screened is ultimately up to the individual and that the health care provider should acknowledge the patient's decision to be screened or not.

## Prevention and treatment of cervical cancer

Almost all participants perceived that cervical cancer is incurable and assumed that it cannot be prevented; however, the participants believed that they were not vulnerable to this disease and that the screening was not necessary. One participant mentioned that, *"If we go to prevention without any symptoms, what is the point? People might assume [*those seeing us visiting the screening room*] as we have cancer. The other issue is that if people know that we have cancer, we will be stigmatized, so better not to get known or get treatment"* (42 years old female, Boloso Sore).

In FGDs, another point highlighted by men participants was that the health facilities were not working toward the prevention of cancer but were instead providing contraception to their wives. One participant stated, *"We don't want our wives to go to a health facility because they usually go for contraceptives, although we want to continue having children. It sounds as though health professionals are promoting contraception a lot and females ended up not wanting to have children"* (32 years old male, Sodo Zuria).

No participant was aware of the treatment of cervical cancer, and everyone perceived this disease to be incurable. One participant clarified that they had never heard that any cancer can be treated.

At the end of each FGD and IDI, participants were asked if they would attend health education sessions about cervical cancer prevention. All expressed high interest learning about the disease but preferred any other free method of screening procedures.

## Discussions

This study explored communities' awareness of cervical cancer, its risk factors, symptoms, prevention, and treatment among rural men and women of Wolaita zone, southern Ethiopia. The country is one of many other low and middle-income countries with a high incidence of cervical cancer that have been unable to establish or enhance appropriate and effective cervical cancer prevention programs. Thus, the findings of our study may add to a small body of published articles and provide a significant opportunity for intervention about cervical cancer prevention and treatment in Ethiopia.

Our finding that the participants had not even heard of cervical cancer is consistent with previous reports from Ethiopia [18], Ghana [19] and Korea [20]. In a study in Ethiopia [18], majority of the participants were unaware of cervical cancer and HPV infection as its risk factor and believed that they were not susceptible to this disease. In another qualitative study assessing the awareness of cervical cancer amongst males in Korea [20], the participants' degree of awareness of this disease was low and their concern and knowledge in this regard were poor. Likewise, a number of studies have reported the lack of awareness of cervical cancer as an important barrier to participation in screening [7, 21, 22]. In developing countries, a lack of knowledge regarding cervical cancer is most likely linked to poor education and low socio-economic status. A study conducted in a developed country, Virginia USA, on the other hand, shows that participants were able to identify the screening test associated with cervical cancer and friends and families were their sources of information [23]. Further, a recent study showed that in both developing and developed countries, women with a poor socio-economic status were at a high risk of developing cervical cancer but their rate of participation in screening remained low [24]. The study demonstrated that the rate of availing cancer screening depended on the socio-economic status, and this bias was greater in countries that did not offer population-based cancer screening program. These reports indicate that participation in cervical cancer screening is linked to income and education. Specifically, uneducated women with poor socio-economic status tend to show low screening participation.

In our study, womb expulsion due to prolonged labour during childbirth, as a result of failure for not going to the health institution for delivery, was considered by our study participants as a cause of cervical cancer. Our finding agrees with a recent qualitative study conducted in Ethiopia [25], in which the study participants agreed, though not described in detail, that cervical cancer can be caused when a woman gives birth at home. This clearly reflects the need for public information campaign by trusted sources, including health workers, the media, and other stakeholders at the grassroots.

Our finding that pelvic examination is a barrier to screening corroborates previous reports from Minnesota [26], Guatemala [27], China [28] and Bangladesh [29]. In a qualitative study in Minnesota [26], pelvic examination was perceived as invasive and the use of speculum was considered problematic. Moreover, our findings are consistent with reports from Kenya, suggesting that the fear of undergoing a pelvic examination as well as the fear of disease and death impede screening participation [7]. Given the cultural identity and age of the study participants, this finding was not surprising. Nonetheless, as long as performed in an environment with sufficient privacy, speculum examination was perceived by adolescent girls as a 'modern approach'.

Furthermore, a recent publication from World Health Organization reported that HPV self-sampling offers an additional option to improve cervical cancer screening coverage by creating a more comfortable atmosphere for the patient [30]. A study in Bangladesh has emphasized the availability of female healthcare providers and privacy as a requirement for gynecological examination [29]; In that study, speculum examination was considered

acceptable by both women and men as long as it was performed by a female health care provider in an environment with adequate privacy.

Regarding perceptions about cervical cancer screening, our finding shows that most of the study participants explained that they only go to the hospitals or health institutions when they have symptoms and apparent illnesses, including abdominal pain and headaches. A similar finding from Nepal showed that study participants did not feel any need to seek health care unless they experienced symptoms of some kind [21]. Further, our finding is in line with one from a study conducted in Uganda. The investigators found that study participants who had screened for cervical cancer mostly had had its sign and symptom [6]. This might be due to a high level of illiteracy, dependency on traditional practices, poor socio-economic conditions, and poor health awareness.

## Implications for research, policy-making and practice

Healthcare providers working with women, particularly older women, should ensure that the information they provide is culturally sensitive and should acknowledge the cultural beliefs of these groups of the population. Further, awareness about cervical cancer and participation in cancer screening can be addressed through culturally relevant health education and communication interventions.

In this study, lack of awareness about the symptoms of cervical cancer combined with its sexual risk factors and association with HIV led the study participants to believe the condition to be the kind of illness that HIV-infected women would get. This leads to the stigmatization of patients with cervical cancer. Therefore, the development of community-based participatory approach stigma reduction interventions, through improving access to information and education, is paramount to the local communities. Moreover, a large qualitative study exploring the prevalence of cervical cancer-related stigma and its association with health outcomes may provide valuable lessons on how to better respond to cervical cancer-related stigma.

Our finding on the participants' preferences of cervical cancer screening procedures provides initial evidence for researchers to further explore the satisfaction and acceptance of various screening methods. Additionally, based on our findings, important factors that dissuade women from getting screened for cervical cancer are the understanding of the concept of disease prevention, experience of the screening procedure, awareness of the availability of cervical cancer prevention services, and attitude towards cancer in general. Therefore, interventions focusing on strengthening the existing services and increasing awareness regarding the importance of cervical cancer screening prior to the onset of signs and symptoms amongst communities are critical initial steps to establish an effective cervical cancer prevention campaign. Moreover, trained community healthcare workers can be engaged for encouraging women to utilize the screening services. In addition, information and education plans through printed materials as well as facility-, community- and media-based approaches can promote service utilization. Furthermore, such activities can make people aware of the fact that with appropriate treatment, cervical cancer can often be cured. In this regard, lessons can be learned from a study conducted in Ghana [31], which reported a comprehensive community-based educational campaign on cervical cancer and screening focusing on the cause, risk factors, signs and symptoms, complications and prevention methods. This community-based education used lectures, discussions, videos, and leaflets and has been evidenced by facilitating supportive change and improved knowledge and perception about cervical cancer and screening. Thus the study finding can be used as a blueprint to craft a successful community awareness program regarding cervical cancer screening, prevention, and treatment.

## Conclusion

This study indicates rural communities in the zone had limited knowledge about cervical cancer and even less about risk factors, screening, treatment, and prevention. Factors such as lack of awareness, fear of pelvic exam, having many contending issues, a belief that cervical cancer is not curable, fear of having a positive result, and the stigmatization of cervical cancer are also key barriers to engaging in cervical cancer screening programs. Furthermore, a comprehensive education and information intervention focusing on cervical cancer-related risk factors, signs and symptoms, treatment, complications, and prevention methods is critical in improving communities' perceptions of the disease.

## Supporting information

**S1 File. Supplementary file.**
(DOCX)

## Acknowledgments

We would like to express our heartfelt gratitude to Wolaita Sodo University for providing fund to conduct this research. We also have a special thank for Mrs. Biruktawit Alemu Wolde, Mr. Yohannes Gebeyehu Guta and Mrs. Bekalwa Teshome Amare for their critical review of the manuscript.

## Author Contributions

**Conceptualization:** Birhanu Wondimeneh Demissie, Gedion Asnake Azeze.

**Data curation:** Birhanu Wondimeneh Demissie, Gedion Asnake Azeze, Netsanet Abera Asseffa.

**Formal analysis:** Birhanu Wondimeneh Demissie, Gedion Asnake Azeze, Netsanet Abera Asseffa, Mohammed Suleiman Obsa.

**Funding acquisition:** Netsanet Abera Asseffa, Eyasu Alem Lake.

**Investigation:** Birhanu Wondimeneh Demissie, Gedion Asnake Azeze, Taklu Marama Mokonnon.

**Methodology:** Birhanu Wondimeneh Demissie, Gedion Asnake Azeze, Eyasu Alem Lake, Befekadu Bekele Besha, Kelemu Abebe Gelaw, Taklu Marama Mokonnon, Natnael Atnafu Gebeyehu.

**Project administration:** Birhanu Wondimeneh Demissie, Gedion Asnake Azeze.

**Software:** Netsanet Abera Asseffa, Befekadu Bekele Besha.

**Supervision:** Birhanu Wondimeneh Demissie, Gedion Asnake Azeze, Kelemu Abebe Gelaw, Natnael Atnafu Gebeyehu, Mohammed Suleiman Obsa.

**Validation:** Birhanu Wondimeneh Demissie, Gedion Asnake Azeze.

**Writing – original draft:** Birhanu Wondimeneh Demissie, Gedion Asnake Azeze.

**Writing – review & editing:** Netsanet Abera Asseffa, Eyasu Alem Lake, Befekadu Bekele Besha, Kelemu Abebe Gelaw, Taklu Marama Mokonnon, Natnael Atnafu Gebeyehu, Mohammed Suleiman Obsa.

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
