## [Decision Letter · Decision Letter 0]

11 Aug 2021

PONE-D-21-21479

Communities’ perception of cervical cancer and screening in Wolaita zone, southern Ethiopia: A qualitative study

PLOS ONE

Dear Dr. Gedion Asnake Azeze,

Thank you for submitting your manuscript to PLOS ONE. After careful consideration, we feel that it has merit but does not fully meet PLOS ONE’s publication criteria as it currently stands. Therefore, we invite you to submit a revised version of the manuscript that addresses the points raised during the review process.

Thank you for your submission that covers an important health problem globally, and more so, in the context of Africa. It is well presented but there are key areas for improvement as show by our two reviewers. Ensure you provide the missing details in the methodology. Remember to show findings stratified by key dimensions including sites, age, and gender. Stregthten implications of your findings broadly, but more specifically for the community of research.

We look forward to receiving your revised manuscript.

Kind regards,

Violet Naanyu, PhD

Academic Editor

PLOS ONE

Journal Requirements:

2. When reporting the results of qualitative research, we suggest consulting the COREQ guidelines: http://intqhc.oxfordjournals.org/content/19/6/349. In this case, please consider including more information on the number of interviewers, their training and characteristics. Moreover, please provide the interview guide used as a Supplementary File.

3. We note that your paper includes detailed descriptions of individual patients/participants. As per the PLOS ONE policy (http://journals.plos.org/plosone/s/submission-guidelines#loc-human-subjects-research) on papers that include identifying, or potentially identifying, information, the individual(s) or parent(s)/guardian(s) must be informed of the terms of the PLOS open-access (CC-BY) license and provide specific permission for publication of these details under the terms of this license. Please download the Consent Form for Publication in a PLOS Journal (http://journals.plos.org/plosone/s/file?id=8ce6/plos-consent-form-english.pdf). The signed consent form should not be submitted with the manuscript, but should be securely filed in the individual's case notes. Please amend the methods section and ethics statement of the manuscript to explicitly state that the patient/participant has provided consent for publication: “The individual in this manuscript has given written informed consent (as outlined in PLOS consent form) to publish these case details.

Reviewers' comments:

Reviewer's Responses to Questions

**Comments to the Author**

1. Is the manuscript technically sound, and do the data support the conclusions?

Reviewer #1: Yes

Reviewer #2: Partly

2. Has the statistical analysis been performed appropriately and rigorously? 

Reviewer #1: N/A

Reviewer #2: N/A

3. Have the authors made all data underlying the findings in their manuscript fully available?

Reviewer #1: Yes

Reviewer #2: Yes

4. Is the manuscript presented in an intelligible fashion and written in standard English?

Reviewer #1: Yes

Reviewer #2: Yes

5. Review Comments to the Author

Reviewer #1: While I find the manuscript technically sound, and  it has a lot of data - including verbatim reports, I found it a little shallow because it is merely descriptive. The knowledge it generates (that rural communities in the context of study do not have adequate information on the cancer of the cervix and therefore there is need to educate them) is almost obvious. Moreover, I have some concerns:

i) Would the author consider referring to Communities’ 'perceptions' rather than ‘perception’? I am sure that there are many perceptions -and the author presents many of these on different aspects of cervical cancer and screening in Ethiopia. In fact, if consideration was of only one aspect such as cause of cancer, there would be many perceptions within one community - likewise in many communities.

ii) Would the author consider referring to researchers/research assistants instead of 'collectors'? In qualitative data collection, one would not merely be a data collector.

iii) I am curious – were interviews done in a local language? What is the local term for ‘cervical cancer’? In many African communities, there is no word for cancer of the cervix per se. For many people it would be cancer of the stomach. How did the interviewer introduce or describe it?

iv) There is reference to HIV/AIDS as a disease. It is important to distinguish between HIV and AIDS. Again what is HIV/AIDS in the local language?

The manuscript is derived from a qualitative descriptive study design so statistical analysis does not apply.

The presentation is intelligible but like I allude in a previous comment, there is little novelty.

Reviewer #2: REVIEW REPORT: Communities’ perception of cervical cancer and screening in Wolaita zone, southern Ethiopia: A qualitative study

The paper is a qualitative study entitled “communities ‘perception of cervical cancer and screening in Wolaita zone Southern Ethiopia “. This title is quite clear and precise. The abstract is well detailed and addresses the main research question. The study addressed one main objective namely; to explore communities’ perception of cervical cancer and screening among women in Wolaita zone, southern Ethiopia. The manuscript is an excellent demonstration of views and attitudes of a rural area population towards cervical cancer and screening. In their paper the authors used Focus Group Discussions (FGDs) and In Depth Interviews (IDI) to collect data from 59 participants living in Southern Ethiopia. Data collected was on knowledge about cervical cancer, its symptoms perception and attitude towards screening. The study was conducted in two urban and two rural Kebeles –which is the lowest administrative unit in Ethiopia. The sample of the study comprised of non-health professionals who had not had any training on cervical cancer residing in the study area. A proportional number of FGDs and IDIs were held in the two selected districts. Data was qualitatively presented.

The results of the study revealed that most participants had not heard about cervical cancer but had knowledge on cancer in general. Though cervical cancer was perceived to be incurable most participants assumed that they were not vulnerable to the disease and therefore screening was not necessary.

The authors conclude that rural communities in the zone had limited knowledge about cervical cancer and even less about risk factors, screening, treatment and prevention and therefore recommended that there is a great need for cancer education and prevention in Ethiopia.

The strength of this manuscript is that it addresses a pertinent issues concerning cervical cancer, presents findings and as such the paper represent a community’s view which will influence policy in Ethiopia and generally increase knowledge on cervical cancer screening in Sub Saharan Africa. This manuscript does an excellent job in researching on cervical cancer among non -medical/ health participants. Though an issue affecting women, the authors did an excellent job to include men in their study, this brings out a community’s holistic view on cervical cancer. Cervical cancer being a threat in sub-Saharan Africa with screening uptake being very low, this study is very relevant in highlighting some of the barriers to cervical cancer screening. The study is therefore quite relevant and is an attempt in filling the research gap as far as cervical cancer and screening are concerned.

The paper however fails to discuss the interpretation and implications of the findings

Secondly, it does not strongly bring out the new knowledge bone from the research.

While the methodology section provides some information, it is still unclear whether the authors used the same participants for both FGDs and IDI, and if so, what new findings each provided.

No differences were brought out concerning the general findings by gender and age, yet the authors had stratified the population based on age and gender.

Based on the findings I state that though the topic is quite relevant, it is important for the authors to address the key concerns raised so as to make their work more useful to readers. I am available to review a corrected manuscript. Other comments have been uploaded.

6. PLOS authors have the option to publish the peer review history of their article (what does this mean?). If published, this will include your full peer review and any attached files.

Reviewer #1: No

Reviewer #2: No

---

## [Author Response · Author response to Decision Letter 0]

26 Aug 2021

26/08/2021

Violet Naanyu, PhD

Academic Editor

PLOS ONE

RE: PONE-D-21-21479: Communities’ perception of cervical cancer and screening in Wolaita zone, southern Ethiopia: A qualitative study

Dear Dr Naanyu, 

Thank you for considering our manuscript and for arranging for it to be reviewed by two reviewers. We have tried to address your comments and the comments / suggestions from the two reviewers. 

Please find for your kind consideration the following:

In the Response to Reviewers, we copy each of the comments / suggestions and provide the RESPONSE underneath (below pages 2-13). 

We also provide a marked-up copy of the manuscript that highlights changes made to the original version and this is uploaded as a separate file labelled “Revised Manuscript with Track Changes”. 

Finally, we provide an unmarked version of the revised manuscript without tracked changes and this is uploaded as a separate file labelled 'Manuscript'. 

We have been carefully through the peer review and have revised our paper accordingly. We feel that the paper is much improved as a result of this peer review process, and thank you for taking it to this stage. 

While hoping that these changes would meet with your favourable consideration, we hold ourselves at your entire disposition for any further information or other changes you might require.

Best wishes 

Gedion Asnake Azeze; on behalf of the co-authors

POINT BY POINT RESPONSE TO THE EDITOR AND REVIEWERS SUGGESTIONS

Response to Academic Editor: 

Ensure you provide the missing details in the methodology. 

RESPONSE: Given the general nature of the comment, we have tried to add few more details in methods sections. 

Remember to show findings stratified by key dimensions including sites, age, and gender.

RESPONSE: Changes were made to all quotes at which sites, age and sex dimensions were indicated.

Stregthten implications of your findings broadly, but more specifically for the community of research. 

RESPONSE: we have tried to add more on ‘Implication to practice section’ and for that matter; we have modified this section to ‘Implication for research, policy and practice’

RESPONSE: We have checked and have ensured that the manuscript meets the journal’s requirements. 

When reporting the results of qualitative research, we suggest consulting the COREQ guidelines: In this case, please consider including more information on the number of interviewers, their training and characteristics. Moreover, please provide the interview guide used as a Supplementary File.

RESPONSE: we have made changes and guide has been added as ‘Supplementary file_1’

We note that your paper includes detailed descriptions of individual patients/participants. As per the PLOS ONE policy on papers that include identifying, or potentially identifying, information, the individual(s) or parent(s)/guardian(s) must be informed of the terms of the PLOS open-access (CC-BY) license and provide specific permission for publication of these details under the terms of this license. Please download the Consent Form for Publication in a PLOS Journal. The signed consent form should not be submitted with the manuscript, but should be securely filed in the individual's case notes. Please amend the methods section and ethics statement of the manuscript to explicitly state that the patient/participant has provided consent for publication: “The individual in this manuscript has given written informed consent (as outlined in PLOS consent form) to publish these case details.. 

RESPONSE: we have made all the necessary amendments.

Response to Reviewer 1: 

We thank the reviewer for reviewing this paper and for the detailed comments and suggestions.

The different points raised by the reviewer include:

While I find the manuscript technically sound, and it has a lot of data - including verbatim reports, I found it a little shallow because it is merely descriptive. The knowledge it generates (that rural communities in the context of study do not have adequate information on the cancer of the cervix and therefore there is need to educate them) is almost obvious. 

RESPONSE: we would like to thank the reviewer for encouraging response. By taking this concern of the reviewer and the comments from reviewer#2 on the method, result, discussion and implication section of the manuscript, we have made several amendments and we believe these changes have been useful in improving this paper.

i) Would the author consider referring to Communities’ 'perceptions' rather than ‘perception’? I am sure that there are many perceptions -and the author presents many of these on different aspects of cervical cancer and screening in Ethiopia. In fact, if consideration was of only one aspect such as cause of cancer, there would be many perceptions within one community - likewise in many communities.

RESPONSE: we found this comment very helpful and we have replaced the word ‘perception’ to ‘perceptions’ throughout the document.

ii) Would the author consider referring to researchers/research assistants instead of 'collectors'? In qualitative data collection, one would not merely be a data collector.

RESPONSE: we have replaced data ‘collectors’ by ‘research assistants’. 

iii) I am curious – were interviews done in a local language? What is the local term for ‘cervical cancer’? In many African communities, there is no word for cancer of the cervix per se. For many people it would be cancer of the stomach. How did the interviewer introduce or describe it?

RESPONSE: interviews were conducted in local language, ‘Wolaitegna’ and ‘Amharic’ language; the local term for ‘cervical cancer’ is “Barka” in Wolaitegna and “Yemahitsen ber nekersa” in Amharic (line# 193-194).

iv) There is reference to HIV/AIDS as a disease. It is important to distinguish between HIV and AIDS. Again what is HIV/AIDS in the local language?

RESPONSE: we have made corrections on HIV/AIDS. There is no local term for HIV/AIDS and study participants were communicating throughout the discussion mentioning HIV/AIDS as it stands. 

The manuscript is derived from a qualitative descriptive study design so statistical analysis does not apply. The presentation is intelligible but like I allude in a previous comment, there is little novelty.

RESPONSE: thank you but we now assume that the manuscript will get some improvements after incorporating and addressing the suggestions and comments.

Response to Reviewer 2: 

We thank the reviewer for reviewing this paper and for the detailed comments and suggestions. 

We have tried to revise the manuscript in line with these comments and suggestions.

The different points raised include:

The strength of this manuscript is that it addresses a pertinent issues concerning cervical cancer, presents findings and as such the paper represent a community’s view which will influence policy in Ethiopia and generally increase knowledge on cervical cancer screening in Sub Saharan Africa. This manuscript does an excellent job in researching on cervical cancer among non -medical/ health participants. Though an issue affecting women, the authors did an excellent job to include men in their study, this brings out a community’s holistic view on cervical cancer. Cervical cancer being a threat in sub-Saharan Africa with screening uptake being very low, this study is very relevant in highlighting some of the barriers to cervical cancer screening. The study is therefore quite relevant and is an attempt in filling the research gap as far as cervical cancer and screening are concerned. The paper however fails to discuss the interpretation and implications of the findings and secondly it does not strongly bring out the new knowledge bone from the research.

RESPONSE: Thank you for the encouraging comment. We take note of your concern about discussing the interpretation and implications of the findings and we have made several amendments in the revised manuscript. 

While the methodology section provides some information, it is still unclear whether the authors used the same participants for both FGDs and IDI and if so what new findings did each provide.

RESPONSE: No, they were exclusive, none of FGD participants included in IDI and vice-versa.

No differences were brought out concerning the general findings between gender and age yet the authors had stratified the population based on age and gender. 

RESPONSE: The general finding is not different but it was stratified to help readers know the details of the participants.

Based on the findings I state that though the topic is quite relevant, it is important for the authors to address the key concerns raised so as to attract wider readability after publication.

RESPONSE: thank you. We found this advice very relevant to improve our manuscript and we have tried to address the concerns raised by both authors by adding new sentences, paragraphs and references.

2. Discussion of specific areas for improvement

A. Major 

The scope of the study is well provided and the presentation is consistent throughout the manuscript. Though the manuscript is well written and nicely presented descriptively, the area of the study setting is well described and brings out the necessity for this kind of study. Though this manuscript presents a balanced assessment of cervical cancer and shows evidence of being well researched, the authors were mean with pertinent information and therefore the following areas need some revision;

• The scope of the study is well provided and the presentation is consistent throughout the manuscript. Though the manuscript is well written and nicely prese The findings that the study produces are very critical but have no implications, and they only need one or two sentences to bring out the implication of each findings.

RESPONSE: Thank you. We take note of your advice about findings implications and we have made changes accordingly.

• The manuscript is well written with key subtopics availed. The authors however need to be more analytical in data presentation. The excerpts are presented and not interpreted at all. Making it difficult to make exact meaning from the excerpts. It is always good practice to interpret all findings, this makes it easy to come up with the implications of the findings. The manuscript would be more useful to a broader readership if the authors moved from just providing results to interpreting them and giving the implication as well.

RESPONSE: thank you once again for your valuable comments. Since the reviewer provided specific areas for improvements hereunder (especially on the result, discussion and Implication section), we’ve tried to interpret each finding and to put their implication. 

• I tend to think that the authors oversimplified the findings thus inhibiting the deeper message that the subject under study could provide. 

RESPONSE: for this comment, and including the previous two, we have tried to modify the ‘result’ and ‘discussion’ sections and we have broadened the ‘Implications for practice’ section by adding some valuable implication statements that summarize our findings.in the revised manuscript, this section has been changed to ‘Implication for research, policy and practice’.

• The citations should also have deeper analysis rather than simply being quoted, they should be linked to the study either as an agreement in findings or a deviation from the findings. 

RESPONSE: we have tried to discuss in detail on some of cited references. From Line 302 to 308, for instance, there are three cited references. We have mentioned that the findings of these studies were in agreement with ours. So, from these three citations, we picked two and we tried to discuss in detail (line 303 to 308). Similarly, we followed the same thing on line# 315–319; line# 322–326 and line# 328–333. 

• The manuscript provides a recommendation but has no summary on suggested direction that people in the study area need to take and as such there is no takeaway. 

RESPONSE: After this comment, we have tried to put some directions on our recommendations particularly on ‘Implication for research, policy and practice’ section. 

Background

• Line 46- What is the source of this definition? 

RESPONSE: We put a citation on the mentioned definition (line# 48)

• Line 50- Any woman who is sexually active is at risk of getting HPV- this needs to be explained further so that it does not appear hanging. 

RESPONSE: now, we believe that the statement is well explained (line# 51-53 )

• Some brief literature review of a country where cervical cancer is low could improve the background.

RESPONSE: we have made a literature search and we added a new statements that describes low cervical cancer prevalence (line# 60-64)

Sampling size and sampling technique

• How did you get the sample? What was the inclusion and exclusion criteria? 

RESPONSE we have modified both the ‘participants’ and ‘sample size and sampling technique’ sections; adding inclusion and exclusion criteria on ‘participants’ section and sampling method on ‘sample size and sampling technique’ (line# 108-112)

• Line 86-state why the two districts were selected to participate in this study ( Boloso Sore & Sodo Zuriya). 

RESPONSE: the two districts, namely Boloso Sore & Sodo Zuriya of Wolaita zone, were selected randomly (line# 113) from the total of 15 districts of Wolaita zone.

• Line 110-112 – 4 females and 2 male nurses were trained- are they research assistants? If so explain why they had all to be nurses to participate in data collection.¬¬

RESPONSE: Yes, those 4 females and 2 male nurses were our research assistants. These research assistants were fluent in the required local languages; they were used to prompt discussion and elicit further details through probes during FGDs. We have replaced words like ‘collectors’ or ‘data collectors’ with ‘research assistants’ throughout the document because this comment was raised by reviewer#1. They were all nurses because of their previous qualitative data collection experiences (line# 128-132).

• Line 136- which themes were they- mention the themes here to complete this statement. 

RESPONSE: thank you. We have mentioned the themes and completed the statement.

• Line 153-154 Awareness about cervical cancer- authors talk of ‘most participants’-though this is a qualitative study, it does no wrong to mention roughly how many participants instead of using the term “most”.

RESPONSE: We understand the reviewer’s point. Nevertheless, we find this comment a little difficult to address since we did not take a note on the exact number of participants who have not heard about cervical cancer. During the analysis, when a term or a concept gets repeated several times by different group of participant we take that as most indicating that it is more of their conclusion or what most assume to be true.

• Line 158- Authors talk about misconceptions but only present one misconception from the FGDs and not IDIs do the authors imply that all the respondents had similar misconceptions? Secondly, what is the implication of these misconceptions? 

RESPONSE: Thank you. First, since we have got only one misconception from FGDs, we made a correction on the word ‘misconceptions’ and we replaced it with ‘misconception’. Not all participants but most of them had the mentioned misconception. Secondly, since this misconception is associated with lack of awareness on what cervical cancer is, we have tried to put the implication (together with other awareness related issues) on ‘Implication for research, policy and practice’ section (line# 378-9). 

Data collection procedures

• It is not clear whether they are the same participants who were in the FGDs and interviews? Explain to clarify this. If they were different how did you choose those to participate in the FGD and who to participate in the IDI? If they participated in both what new information were you looking for and how did you carry out this? How many participants did you get per area? 

RESPONSE: No, participants who participated in FGDs and IDIs were not the same. We used the same criteria to select participants. A total of eight participants were interviewed and there were fifty-one participants in FGD, each FGD has 5–7 participants.

• L118- authors have used the term “Unapproachable’’ I advise that they get a better term that is not judgmental. 

RESPONSE: changes made as suggested (line# 138). 

• Line 120-121-Make it clearer- which collectors? Are they data collectors? - A bit confusing. Did the data collection exercise take place concurrently in both districts or did the same researchers finish work in one district before going to another. 

RESPONSE: as explained earlier, we have replaced ‘data collectors’ or ‘collectors’ with the phrase ‘research assistants’ in the current revised manuscript after a comment given by reviewr#1. They were research assistants for our study. The same research assistants participated in both districts after finishing the other. 

Data management

• Line 127-129- which quality control procedures did the authors establish for their study? 

RESPONSE: the quality control procedure for the study that we followed include: recruiting research assistants with qualitative study experience, training provided for the research assistants and pre-testing the instrument. 

Results

• Table 1 – needs some write up to explain the figures rather than simply putting the table and leaving it that way, 

RESPONSE: we have introduced a sentence and we have now tried to describe the table. Since there are limited variables in the table, we failed to add more description. 

• It is important to state how many participants were in IDIs

RESPONSE: In methods section (line# 119), we have indicated number of IDIs and each IDI had one participant. We have not had any group interviews.

• Did all the 59 participate in IDIs?

RESPONSE: No, IDI and FGD had different participants.

Line 166- 183 Risk factors for cervical cancer

• This section is well explained though again the urban responses are not distinguished from the rural areas yet such distinctions can be quite important for recommendations

RESPONSE: Although the participant mentioned of rural, it is worth to note that she is from urban setting. The thing is little to no difference was observed among the residents of urban and rural, indicating that there is poor awareness from both sides. Hence, the overall recommendation would work for both settings.

• Line 181-183 talks about the men FGDs what came out from the women FGDs? This would also bring out gender differences as far as views on cervical cancer are concerned thus enriching the study. 

RESPONSE: we have made changes accordingly after reviewing the transcriptions and code report (line# 212).

• The authors had also mentioned earlier that younger men/women were in different FGDs- what was the difference in the findings based on age?

RESPONSE: no difference was observed based on their any of socio-demographic characteristics, the detailed descriptions were just to aid readers and to further contextualize in any future research.

Line 184-200-Awareness about symptoms of cervical cancer 

• What are the implications of these awareness of the symptoms?

RESPONSE: we have added on the implications of poor or lack of awareness on the symptoms of cervical cancer (line# 360-367; 377-384).

Line 201-Perception about cervical cancer screening

• This section should bring out differences based on gender age and location (urban & rural) and also between the two districts of Wolaita zone- namely Boloso sore & Sodo Zuriya as well as the two urban and two rural Kebeles to strengthen the findings. If the findings are similar in these regions, what is the implication? What if they are different? Do the authors imply that all the 9 FGDs and 8 IDIs had similar findings? 

RESPONSE: like we mentioned earlier, it has no difference based on age, sex, place of residence or any other characteristics, however descriptions were reported to aid readers. If this is found to be unnecessary do let us know the reason then we shall remove it. 

Line 229- prevention and treatment of cervical cancer

Let the authors bring out the key points concerning cervical cancer prevention as discussed by 

i) The urban dwellers

ii) Rural dwellers

iii) Young girls

iv) Young men

v) Older men/women

RESPONSE: the analysis showed that there is no major difference observed, quotes were picked up when found to be more expressive but the overall summary before quotes shows the perceptions of majority of the participants. Hence, classification based on the above guide was not found to be necessary.

• What are the implications of the same to treatment? Even if it is a qualitative study it needs to be detailed in presentation of results so as to clearly bring out the implications of the findings.

RESPONSE: based on the given comment, attempts were made to make changes

• Though it is a quantitative study, the authors should avoid using terms like “several, many” etc just state how many. And where possible put figures or percentages or eg out of the 4 FGDs, 3 were of the view that e.g cervical cancer is similar to HIV/AIDS.

RESPONSE: Thank you. Changes have been made after reviewing the transcription and code reports.

Risk factors for cervical cancer

• Line 175-180-Interpret the findings and give meaning to what the participants mean by talking about womb expulsion.

RESPONSE: comment was accepted and we have made changes (line# 321–326)

• Line 206-208- what is the implication of such results?

RESPONSE: our finding supports the need for cervical cancer prevention education. In our revised manuscript, we have tried to show the implication of such findings under the section ‘Implication for research, policy and practice’(line# 373-379).

• Interpret the excerpts and give their implication eg line 221-228- looks disjointed when not explained before getting into another section.

RESPONSE: thank you. Together with the comment given below, we have tried to interpret and explain the finding and we have tried to put the implications (271-273; 368-369).

• Line 245- What do the authors mean by “any other free method of screening”? Explain further, is it the financial free method or is it the screening procedure that the participants prefer.

RESPONSE: from our finding one can see that provider initiated pelvic examination was identified as a barrier for screening uptake. Therefore, by saying “any other method of screening”, we mean the preferred screening procedures but not the financial fee. For this, we have tried to put implications (368-369).

Discussion

• Line 262-264 what does this imply based on this study findings?

RESPONSE: Changes have been made (line# 316-320).

• Line 265-266 what was the most common finding of these studies that was common to the authors’ study?

RESPONSE: what makes the study findings similar with that of ours is that having pelvic examination was taken as a barrier for undergoing cervical cancer screening. We have tried to explain by taking and explaining one of these cited reference (line# 329-332).

• Line 264- link this to your study. Did the authors also establish in their study that women of low socio-economic status were at a higher risk of cervical cancer? If so according to their study what is the contributing factor and what new information came from this study?

RESPONSE: Thank you for this valuable comment. We have made changes accordingly (line# 316-320).

• Line 265-Let it be pelvic examination rather than pelvic exam

RESPONSE: we have made a correction.

• Line 274-275- after noting the importance what were the results or the response?

RESPONSE: we have tried to show the results of the mentioned study (line# 341-344).

• Discussion section should be improved by linking the study findings to earlier works on the subject area. 

RESPONSE: we have tried to modify the discussion section based on the given comment.

Minor Issues

• The study needs quick edit, there are a few mistakes on typos, tenses and spellings

RESPONSE: the manuscript was reviewed and we have tried to make corrections

• There are a few minor observations especially on repetition 

RESPONSE: this has been adjusted accordingly

• The study did not have any funding- check financial disclosure section- which states-the authors received no specific funding for this work pg 2 then on pg 22 line 302 - Wolaita Sodo University provided funding for this study, so which is which? Or what is the authors’ interpretation of funding?

RESPONSE: we have added a new section, ’Funding’, and this has been clarified

---

## [Decision Letter · Decision Letter 1]

15 Sep 2021

PONE-D-21-21479R1Communities’ perceptions of cervical cancer and screening in Wolaita zone, southern Ethiopia: A qualitative studyPLOS ONE

Dear Gedion Asnake Azeze, Thank you for submitting your manuscript to PLOS ONE. After careful consideration, we feel that it has merit but does not fully meet PLOS ONE’s publication criteria as it currently stands. Therefore, we invite you to submit a revised version of the manuscript that addresses the points raised during the review process.

Thank you for addressing Reviewer comments.Edit grammatical error and submit a refined copy.

Kind regards,

Violet Naanyu, PhD

Academic Editor

PLOS ONE

Journal Requirements:

Reviewer's Responses to Questions

**Comments to the Author**

1. If the authors have adequately addressed your comments raised in a previous round of review and you feel that this manuscript is now acceptable for publication, you may indicate that here to bypass the “Comments to the Author” section, enter your conflict of interest statement in the “Confidential to Editor” section, and submit your "Accept" recommendation.

Reviewer #1: All comments have been addressed

Reviewer #2: All comments have been addressed

2. Is the manuscript technically sound, and do the data support the conclusions?

Reviewer #1: Yes

Reviewer #2: Yes

3. Has the statistical analysis been performed appropriately and rigorously? 

Reviewer #1: N/A

Reviewer #2: Yes

4. Have the authors made all data underlying the findings in their manuscript fully available?

Reviewer #1: Yes

Reviewer #2: Yes

5. Is the manuscript presented in an intelligible fashion and written in standard English?

Reviewer #1: Yes

Reviewer #2: Yes

6. Review Comments to the Author

Reviewer #1: _ Right from title, the authors should refer to perceptions - It cant be one perception among many communities - well within one community the perceptions are many.

-

Reviewer #2: The authors have addressed all the concerns raised. Just polish up grammatical errors like line 138, line 172-173 and line 244.

7. PLOS authors have the option to publish the peer review history of their article (what does this mean?). If published, this will include your full peer review and any attached files.

Reviewer #1: **Yes: **Eunice Kamaara

Reviewer #2: No

---

## [Author Response · Author response to Decision Letter 1]

22 Sep 2021

22/09/2021

Violet Naanyu, PhD

Academic Editor

PLOS ONE

RE: [PONE-D-21-21479R1]: Communities’ perceptions of cervical cancer and screening in Wolaita zone, southern Ethiopia: A qualitative study

Dear Dr Naanyu, 

Once again, we would like to thank you for considering our manuscript and for arranging for it to be reviewed by reviewer(s). We have tried to address your comments and the comments / suggestions from the reviewer(s). In the ‘Response to Reviewers’, we copy each of the comments and provide the RESPONSE underneath. We also provide a marked-up copy of the manuscript that highlights changes made to the original version and this is uploaded as a separate file called “Manuscript with Track Changes”. Finally, we provide an unmarked version of the revised manuscript without tracked changes and this is uploaded as a separate file labelled 'Manuscript'. 

We have been carefully through the peer review and have revised our paper accordingly. We feel that the paper is much improved as a result of this peer review process, and thank you for taking it to this stage. 

While hoping that these changes would meet with your favourable consideration, we hold ourselves at your entire disposition for any further information or other changes you might require.

Best wishes 

Gedion Asnake Azeze, on behalf of the co-authors

Journal requirement’s: 

Please review your reference list to ensure that it is complete and correct. If you have cited papers that have been retracted, please include the rationale for doing so in the manuscript text, or remove these references and replace them with relevant current references. Any changes to the reference list should be mentioned in the rebuttal letter that accompanies your revised manuscript. If you need to cite a retracted article, indicate the article’s retracted status in the Reference list and also include a citation and full reference for the retraction notice.

Response:

We have reviewed our reference list and we have checked and updated all the URL links.

Response to reviewers

We would like to thank both the reviewer for reviewing this paper and for their comments and suggestions.

Reviewer #1

Right from title, the authors should refer to perceptions – It can be one perception among many communities –well within one community the perceptions are many.

Response:

Thank you. We have addressed this comment during the first revision and we have also checked the consistency in this current revision.

Reviewer #2

The authors have addressed all the concerns raised. Just polish up grammatical errors like line 138, line 172-173 and line 244.

Response:

We have made revisions to correct grammar and language use throughout the document. In this revised manuscript, we also believed we improved the overall tone and readability. We hope that our revisions meet your expectations.

---

## [Editor Report · Decision Letter 2]

17 Dec 2021

Communities’ perceptions towards cervical cancer and its screening in Wolaita zone, southern Ethiopia: A qualitative study

PONE-D-21-21479R2

Dear Gedion Azeze,

We’re pleased to inform you that your manuscript has been judged scientifically suitable for publication and will be formally accepted for publication once it meets all outstanding technical requirements.

Kind regards,

Violet Naanyu, PhD

Academic Editor

PLOS ONE

---

## [Editor Report · Acceptance letter]

31 Dec 2021

PONE-D-21-21479R2 

Communities’ perceptions towards cervical cancer and its screening in Wolaita zone, southern Ethiopia: A qualitative study 

Dear Dr. Azeze:

I'm pleased to inform you that your manuscript has been deemed suitable for publication in PLOS ONE. Congratulations! Your manuscript is now with our production department. 

Kind regards, 

on behalf of

Prof. Violet Naanyu 

Academic Editor

PLOS ONE